

# Sclerostin promotes human dental pulp cells senescence

Yanjing Ou, Yi Zhou, Shanshan Liang and Yining Wang

The State Key Laboratory Breeding Base of Basic Science of Stomatology (Hubei-MOST) and Key Laboratory of Oral Biomedicine Ministry of Education, School and Hospital of Stomatology, Wuhan University, Wuhan, China

## ABSTRACT

**Background:** Senescence-related impairment of proliferation and differentiation limits the use of dental pulp cells for tissue regeneration. Deletion of sclerostin improves the dentinogenesis regeneration, while its role in dental pulp senescence is unclear. We investigated the role of sclerostin in subculture-induced senescence of human dental pulp cells (HDPCs) and in the senescence-related decline of proliferation and odontoblastic differentiation.

**Methods:** Immunohistochemical staining and qRT-PCR analyses were performed to examine the expression pattern of sclerostin in young (20–30-year-old) and senescent (45–80-year-old) dental pulps. HDPCs were serially subcultured until senescence, and the expression of sclerostin was examined by qRT-PCR analysis. HDPCs with sclerostin overexpression and knockdown were constructed to investigate the role of sclerostin in HDPCs senescence and senescence-related impairment of odontoblastic differentiation potential.

**Results:** By immunohistochemistry and qRT-PCR, we found a significantly increased expression level of sclerostin in senescent human dental pulp compared with that of young human dental pulp. Additionally, elevated sclerostin expression was found in subculture-induced senescent HDPCs in vitro. By sclerostin overexpression and knockdown, we found that sclerostin promoted HDPCs senescence-related decline of proliferation and odontoblastic differentiation potential with increased expression of p16, p53 and p21 and downregulation of the Wnt signaling pathway.

**Discussion:** The increased expression of sclerostin is responsible for the decline of proliferation and odontoblastic differentiation potential of HDPCs during cellular senescence. Anti-sclerostin treatment may be beneficial for the maintenance of the proliferation and odontoblastic differentiation potentials of HDPCs.

Corresponding authors
Shanshan Liang,
wb000867@whu.edu.cn
Yining Wang, wang.yn@whu.edu.cn

## INTRODUCTION

Dental caries, trauma, abrasion, attrition, erosion and dental treatment lead to tooth tissues destruction, which eventually results in tooth loss. Non-biological treatment strategies for tooth loss, such as bridges, dentures and implants, may fit poorly or induce foreign body reaction and run the risk of rejection by immune system (*Modino & Sharpe, 2005*; *Yen & Sharpe, 2006*). Dental pulp cells (DPCs), a heterogeneous population of

odontoblasts, epithelia, neurocytes and mesenchymal cells, possess self-renewal and pluripotent differentiation potentials and play a crucial role in maintaining dental pulp homeostasis (*Gronthos et al., 2002*; *Iohara et al., 2004*). Engineering tissue regeneration using DPCs as seed cells is a useful strategy to regenerate the dentin-pulp complex, thereby lengthening the lifespan of teeth by preserving their defensive ability (*Gronthos et al., 2002*; *Karaoz et al., 2011*; *Potdar & Jethmalani, 2015*; *Simon et al., 2011*).

In an aging society, fibrosis, atrophy, loss of cellularity and degeneration of odontoblasts caused by aging make it more difficult to maintain dental health of older individuals (*Nakashima & Iohara, 2014*). DPC senescence, caused by aging or other cytotoxic factors (e.g., oxidative stress due to oral procedures (*Soares et al., 2015*) and irradiation (*Muthna et al., 2010*)), is a state of irreversible cellular arrest accompanied by aggregation of intracellular senescence molecules p16 and p53 (*Rayess, Wang & Srivatsan, 2012*), and changes in secretion of bioactive soluble factors, which lead to the decreased and then the arrest of proliferation and differentiation potentials (*Iohara et al., 2014*). However, little is known about the mechanisms underlying DPC senescence. Previous studies demonstrated that several intracellular factors including p16, p21 and Bmi-1 are involved in the progress of DPC senescence (*Choi et al., 2012*; *Egbuniwe et al., 2011*; *Mehrazarin et al., 2011*). Recently, it was reported that the properties of senescent cells can be reversed by changing the extrinsic microenvironment (*Wagner et al., 2008*; *Nakashima & Iohara, 2014*). Therefore, identifying specific extrinsic factors involved in DPC senescence is of great importance to biologically based tissue regeneration.

Sclerostin is a 190-amino acid secreted glycoprotein encoded by the *SOST* gene. Its expression is restricted to the great arteries (*Zhou et al., 2017*), osteocytes (*Compton & Lee, 2014*), chondrocytes (*Chan et al., 2011*) and cementocytes (*Bao et al., 2013*). The deletion or down-regulation of sclerostin causes high bone mass diseases such as sclerosteosis (*Van Lierop et al., 2011*) and Van Buchem disease (*Loots et al., 2005*). As a well-known negative regulator of bone formation (*Zhang et al., 2016*), sclerostin inhibits the proliferation and differentiation of cementoblasts (*Bao et al., 2013*) and osteoprogenitor cells including osteosarcoma cells (*Zou, Zhang & Li, 2017*) and human mesenchymal stem cells (*Sutherland et al., 2004*). Recently, significantly elevated serum sclerostin was shown in the elderly (*Modder et al., 2011*; *Roforth et al., 2014*), suggesting a possible role of sclerostin in aging-related bone loss as a result of decreased regenerative potential caused by the accumulation of senescent cells (*Farr et al., 2017*). In addition, previous studies demonstrated that the increased sclerostin produced by osteoclasts from aged mice led to reduced bone formation (*Ota et al., 2013*), and anti-sclerostin treatment increased bone mass in aged rats (*Li et al., 2009*). Moreover, the activator of Sirt1, a critical regulator of aging and longevity (*Satoh et al., 2013*), rescued ovariectomy-induced bone loss by decreasing sclerostin expression (*Artsi et al., 2014*). Therefore, it was speculated that sclerostin might impact cellular senescence in terms of differentiation and proliferation.

Similar to the higher level of sclerostin in aged individuals, the expression level of sclerostin varies in embryonic and adult mouse incisors and molars (*Naka & Yokose, 2011*), indicating a possible role of sclerostin in DPC senescence. Additionally,

*Collignon et al. (2017)* found that sclerostin deficiency increased reparative dentinogenesis in mice, implying anti-sclerostin might reverse the decreased regenerative potential of aged dental pulp. Taken together, we hypothesized a possible correlation between sclerostin and DPC senescence. Therefore, the purpose of our study was to elucidate the role of sclerostin in the process of human dental pulp cell (HDPC) senescence as well as aging-related impairment of HDPC proliferation and odontoblastic differentiation.

# MATERIALS AND METHODS

This work was carried out in accordance with the World Medical Association Declaration of Helsinki (*World Medical Association, 2013*). The protocols and procedures were reviewed and approved by the Ethical Committee of the School and Hospital of Stomatology, Wuhan University, China (2015C12).

## Human dental pulp collection

Healthy and fresh human premolars were extracted from 20- to 80-year-old patients who were under orthodontic or periodontal treatment in the Hospital of Stomatology, Wuhan University. All donors gave their informed consent. The teeth were divided into two groups: the young group contained 30 teeth from 20- to 30-year-old patients and the old group contained 20 teeth from 45- to 80-year-old patients.

A total of 10 teeth from each group were used for immunohistochemical analysis, and 10 teeth from each group for qRT-PCR analysis. The remaining 10 teeth from the young group were used for HDPC culture.

## Immunohistochemical analysis

Teeth were fixed with 4% paraformaldehyde at room temperature and decalcified in 10% EDTA solution for more than 6 months. Teeth were cut into five-μm-thick serial sections, which were collected on poly-L-lysine-coated slides. For immunohistochemical staining, the sections were dewaxed and incubated with rabbit anti-sclerostin (1:200; Abclonal, Boston, MA, USA) at 4 °C overnight. The staining was performed using a biotin-streptavidin kit (ZhongShan Biotech, Beijing, China) before the sections were counterstained with hematoxylin.

## Real-time PCR analysis

Total RNA was isolated using the RNAiso kit (Takara, Kusatsu, Japan). First-strand cDNA syntheses were performed by using the PrimeScript™ RT reagent Kit with gDNA Eraser (Takara, Kusatsu, Japan). Real-time polymerase chain reaction for sclerostin, p16, p53, p21, alkaline phosphatase (ALP), osteocalcin (OCN), osteocalcin (OPN), dentin sialophosphoprotein (DSPP) and glyceraldehyde-3-phosphate-dehydrogenase (GAPDH) was performed with the SYBR Green Kit (Takara, Kusatsu, Japan) using an Applied BioSystems 7900HT thermocycler (Applied Biosystems, Foster City, CA, USA). The primers are listed in Table 1. The relative amount or fold change of the target gene was normalized relative to the level of human GAPDH and the control groups.

**Table 1 Primer sequences used for real-time PCR.**

| Genes | Forward (5′–3′) | Reverse (5′-3′) |
| --- | --- | --- |
| GAPDH | AACAGCGACACCCACTCCTC | CATACCAGGAAATGAGCTTGACAA |
| ALP | CGAGATACAAGCACTCCCACTTC | CTGTTCAGCTCGTACTGCATGTC |
| OPN | GCCGAGGTGATAGTGTGGTT | CAACGGGGATGGCCTTGTAT |
| OCN | GGTGCAGCCTTTGTGTCCAA | CCTGAAAGCCGATGTGGTCA |
| DSPP | CAACCATAGAGAAAGCAAACGCG | TTTCTGTTGCCACTGCTGGGAC |
| Sclerostin | TGGCAGGCGTTCAAGAATGA | GCCCGGTTCATGGTCTTGTT |
| P16 | CCCAACGCACCGAATAGTTAC | CAGCAGCTCCGCCACTC |
| P53 | ACCTATGGAAACTACTTCCTGAAA | CTGGCATTCTGGGAGCTTCA |
| P21 | TCAGGGTCGAAAACGGCG | CCTCTTGGAGAAGATCAGCCG |

## Cells and cell culture

Human dental pulp cells were isolated from the healthy dental pulp of 10 premolars from young group. The dental pulp tissues were digested in a solution containing four mg/mL dispase and three mg/mL collagenase type I for 1 h at 37 °C and then cultured in α-modified essential medium (α-MEM; HyClone Laboratories, Inc., Logan, UT, USA) containing 10% fetal bovine serum (FBS; Gibco, Grand Island, NY, USA) at 37 °C under 5% $CO_2$. Confluent monolayers were dissociated with 0.5% (w/v) trypsin-EDTA for subculture. HDPCs were serially subcultured until the cells spontaneously arrested their replication. In brief, $2 \times 10^5$ HDPCs were plated in 60 mm diameter dishes and the medium was changed every 3 days. When the cells reach 80% confluence, all cells were dissociated with 0.5% (w/v) trypsin-EDTA and counted by a cell counting device (Vi-CELL™ XR; Beckman Coulter, Pasadena, CA, USA). The number of population doublings (PDs) was calculated using the following formula: PD = $\log_2$ (cells harvested/ cells seeded) + previous PD (*Huh et al., 2016*; *Kitajima et al., 2011*). For odontoblastic induction, the medium was changed to the odontoblastic induction medium, containing α-MEM, 5% FBS, 100 U/mL penicillin, 100 μg/mL streptomycin, 10 mmol/L β-glycerophosphate, 50 μg/mL ascorbic acid and 10 nmol/L dexamethasone (Sigma-Aldrich Co., St. Louis, MO, USA). Cultures were maintained with a medium change every 3 days.

## Lentivirus packaging and cell infection

Full human *SOST* cds sequence and a human *SOST* sh-RNA were cloned and inserted into the PCDH-CMV-MCS-EF1-copGFP vector (GenePharma, Shanghai, China) and the GV298 vector (Genechem, Shanghai, China), respectively. Lentiviral particles were produced using three-plasmid systems, including pMD2.G, psPAX2 and the individual vectors, with NEOFECT™ DNA transfection reagent (Neofect, Beijing, China) according to the manufacturer's instruction. For infection, HDPCs were incubated with lentiviral particles and polybrene (four μg/mL) in complete medium for 12 h. Cells with successful infection by pCDH-human-SOST were designated SOST, cells infected by sh-SOST were designated sh-SOST, and control cells infected by empty vector were designated Control and sh-Ctrl, respectively. The expression of sclerostin was quantified by real-time PCR and Western blot analysis. The mRNA and protein expression levels were

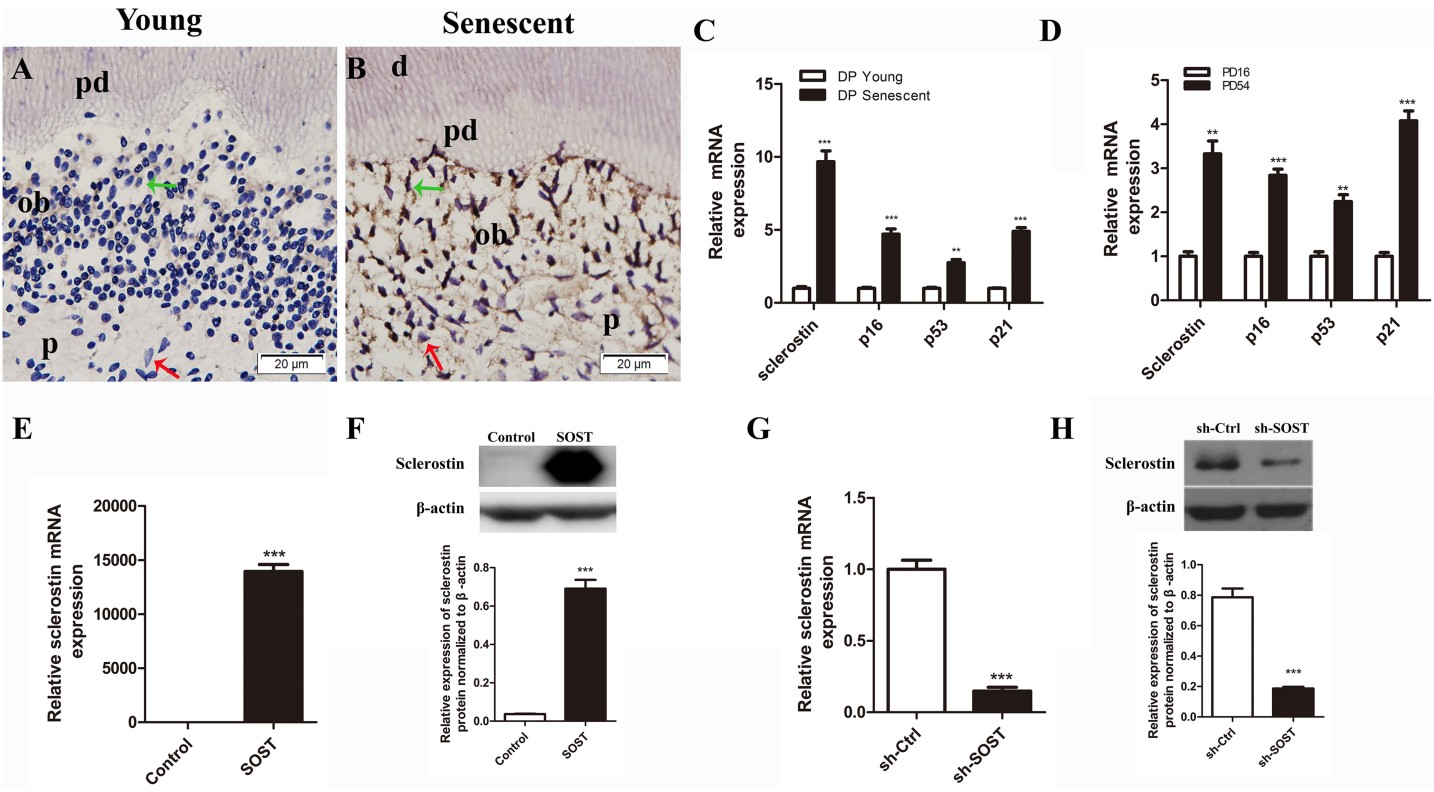

**Figure 1 Expressions of sclerostin in senescent dental pulp, subculture-induced senescent HDPCs and lentiviral infected HDPCs.**
(A) Immunohistochemical staining for sclerostin in young dental pulps; (B) immunohistochemical staining for sclerostin in senescent dental pulps (green arrows point to odontoblasts and red arrows point to dental pulp cells); d, dentin; pd, predentin; p, pulp; ob, odontoblast; scale bar = 20 µm. (C) qRT-PCR analyses of expression levels of sclerostin, p16, p53 and p21 in young and senescent dental pulps; (D) qRT-PCR analyses of expression levels of sclerostin, p16, p53 and p21 in subculture-induced senescent HDPCs. (E) qRT-PCR and (F) Western blot analyses of sclerostin expressions in HDPCs infected with control vector and SOST; (G) qRT-PCR and (H) Western blot analyses of sclerostin expression in HDPCs infected with sh-control vector and sh-SOST. **$P < 0.01$; ***$P < 0.001$.

significantly upregulated in the SOST group ($P < 0.001$, Figs. 1E and 1F), while they were knocked down by over 85% in the sh-SOST group 48 h after infection ($P < 0.0001$, Figs. 1G and 1H).

## Western blot

Cells were lysed in M-PER Mammalian Protein Extraction Reagent (Thermo Fisher Scientific, Rockford, IL, USA) combined with a cocktail of protease inhibitors (Roche Molecular Biochemicals, Rotkreuz, Switzerland). Total lysate was loaded and separated by SDS/PAGE. Primary antibodies specific for sclerostin (1:1,000; Abclonal, Boston, MA, USA), phosphor-β-catenin (p-β-catenin) (1:1,000; Cell Signaling Technology, Danvers, MA, USA) and β-actin (1:5,000; Santa Cruz Biotechnology, Santa Cruz, CA, USA) were used.

## Senescence-associated β-galactosidase assay

To determine senescence of HDPCs, an SA-β-Gal kit (Beyotime, Shanghai, China) was used according to the manufacturer's instructions. In brief, cells seeded on slips were fixed with paraformaldehyde and incubated with senescence-associated β-galactosidase

(SA-β-Gal) overnight. Senescent HDPCs were identified by blue-staining under standard light microscopy.

## Cell proliferation assay

Human dental pulp cells were seeded at a density of $2 \times 10^3$ cells/well in a 96-well plate and cultured for 24 h, and then the medium was replaced with fresh medium. On days 1, 2, 3 and 4, the density of viable cells within each well was quantified with the Cell Counting Kit-8 (CCK-8; Dojindo, Kumamoto, Japan) according to the manufacturer's protocols. The absorbance at 450 nm was measured to calculate the number of viable cells in each well. A well with medium and CCK-8 solution but without cells was used as the baseline.

## Alkaline phosphatase staining and ALP activity assay

Alkaline phosphatase staining was performed according to the manufacturer's instructions (Beyotime, Shanghai, China). For ALP activity, cells were lysed with 0.1% Triton X-100 and 50 μL lysate was mixed with 100 μL p-nitrophenyl phosphate (four mg/mL). The mixture was incubated at 37 °C for 15–20 min. The reaction was stopped by the addition of 0.5N NaOH (100 μL) and read spectrophotometrically at 405 nm. The protein concentration of the lysate was determined as described in the manufacturer's instructions of the Pierce BCA protein assay kit (Thermo Fisher Scientific, Rockford, IL, USA). The enzyme activity was quantified by a p-nitrophenol standard curve and normalized by protein concentration.

## Alizarin red staining

Cells were cultured in 12-well cell culture dishes in the odontoblastic induction medium for 14 days. Then, cells were fixed with 4% paraformaldehyde and stained with 2% alizarin red. The stain was desorbed with 10% cetylpyridinium chloride for 1 h and the absorbance was examined at 562 nm.

## Statistical analysis

All experiments were repeated at least three times. Quantitative results are expressed as the mean ± standard deviation. Data were analyzed by $t$-test and one-way analyses of variance using SPSS 19.0 software (SPSS Inc., Chicago, IL, USA). $P$-values < 0.05 were considered statistically significant.

# RESULTS

## Expression of sclerostin is increased in senescent human dental pulp and subculture-induced senescent HDPCs

The immunohistochemistry assay showed that the expression of sclerostin in young human pulp was very low, and positive sclerostin staining was only found in odontoblasts of young dental pulp. In contrast, strong staining of sclerostin was observed throughout the senescent human dental pulp, especially in the odontoblasts lining near the pre-dentin (Figs. 1A and 1B). Similarly, qRT-PCR analyses showed a higher expression

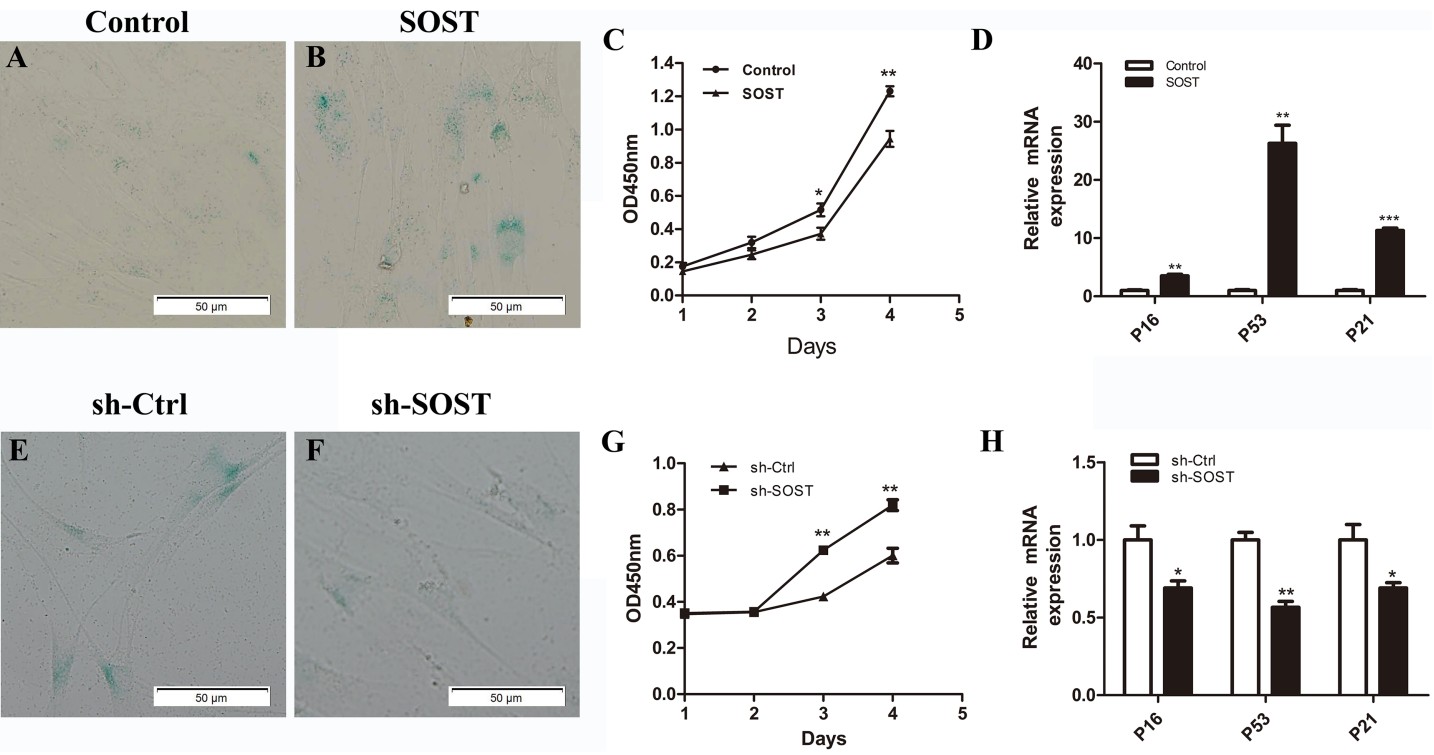

**Figure 2 Effects of sclerostin overexpression and knockdown on senescence and proliferation of HDPCs.** SA-β-Gal staining of Control (A) and SOST (B) HDPCS; (C) cell proliferation activity of Control and SOST HDPCs; (D) qRT-PCR analyses of expression levels of p16, p53 and p21 in Control and SOST HDPCs. SA-β-Gal staining of sh-Ctrl (E) and sh-SOST (F) HDPCs; (G) cell proliferation activity of sh-Ctrl and sh-SOST HDPCs; (H) qRT-PCR analyses of expression levels of p16, p53 and p21 in sh-Ctrl and sh-SOST HDPCs. *$P < 0.05$, **$P < 0.01$; ***$P < 0.001$.

level of sclerostin mRNA in senescent dental pulp, along with increased expression levels of p16, p53 and p21 (Fig. 1C).

Rapidly proliferating HDPCs were serially subcultured until the cells spontaneously arrested replication. HDPCs completing 16 and 54 PDs were defined as young and senescent HDPCs, respectively (*Mehrazarin et al., 2011*). In line with the higher expression levels of sclerostin, p16, p53 and p21 in senescent dental pulp, qRT-PCR analyses showed significantly higher expression levels of sclerostin, p16, p53 and p21 in senescent HDPCs (Fig. 1D).

## Sclerostin induces HDPCs senescence

To access the role of sclerostin in senescence of HDPCs, sclerostin was overexpressed in early-passaged HDPCs. As shown in Fig. 2, sclerostin overexpression significantly increased the number of SA-β-Gal-positive cells. In addition, the CCK-8 assay showed significantly decreased proliferation and higher expression levels of p16, p53 and p21 in sclerostin-overexpressing HDPCs (Figs. 2C and 2D). In contrast, knockdown of sclerostin in late-passaged HDPCs showed fewer SA-β-Gal-positive cells, increased proliferation and decreased p16, p53 and p21 expressions (Figs. 2E–2H). In conclusion, these results suggest that sclerostin induces HDPCs senescence and inhibits HDPCs proliferation via the p16 and p53 signaling pathways.

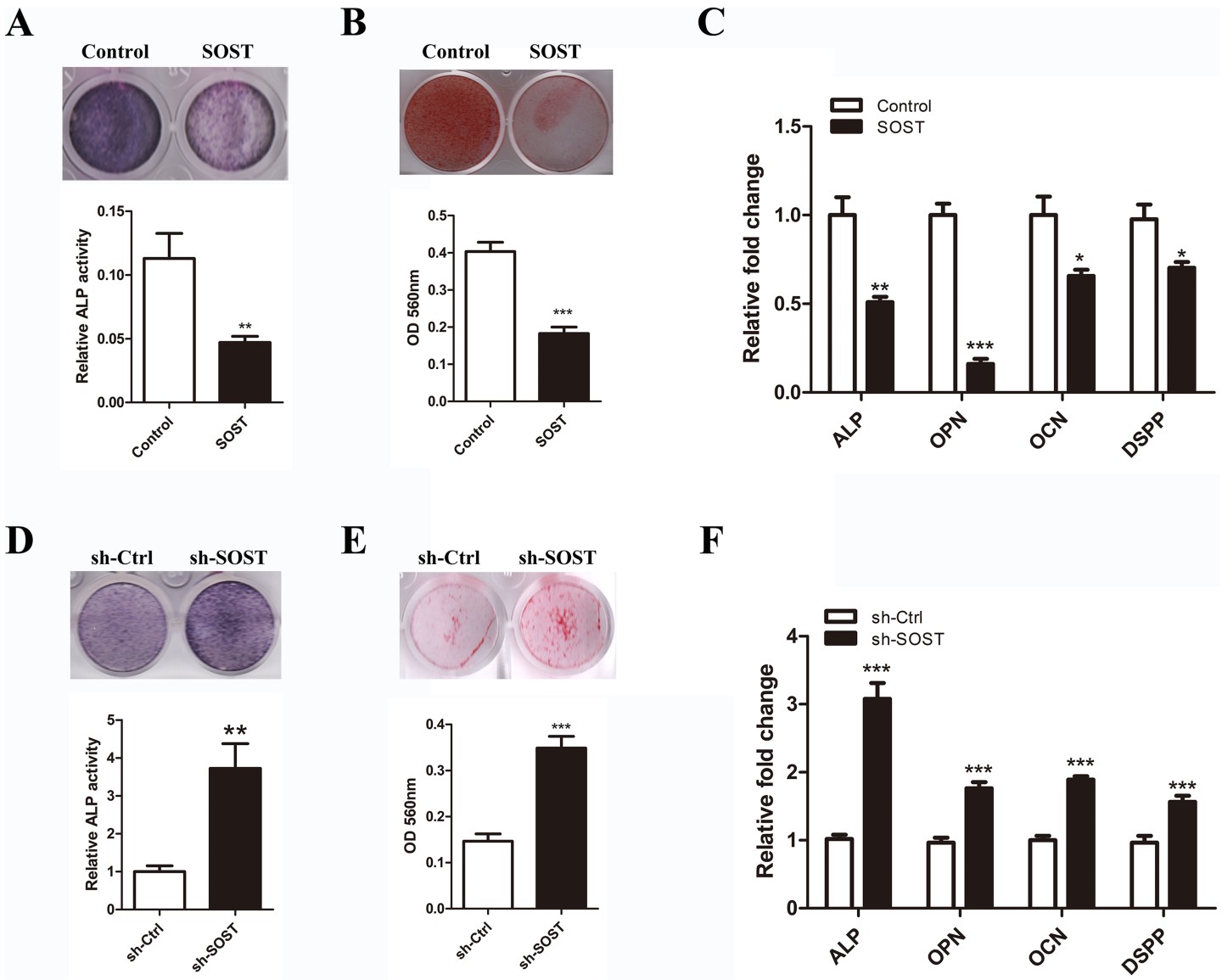

**Figure 3 Effects of sclerostin on odontoblastic differentiation of HDPCs.** (A) ALP staining and ALP activity of early-passaged Control and SOST HDPCs on day 7; (B) alizarin red staining for mineral nodule formation of early-passaged Control and SOST HDPCs on day 14; (C) qRT-PCR analyses of expression levels of odontoblastic markers in early-passaged Control and SOST HDPCs on day 7. (D) ALP staining and ALP activity of late-passaged sh-Ctrl and sh-SOST HDPCs on day 7; (E) alizarin red staining for mineral nodule formation of late-passaged sh-Ctrl and sh-SOST HDPCs on day 14; (F) qRT-PCR analyses of odontoblastic markers in late-passaged sh-Ctrl and sh-SOST HDPCs on day 7. $^*P < 0.05$, $^{**}P < 0.01$; $^{***}P < 0.001$.

## Sclerostin inhibits odontoblastic differentiation of HDPCs

To determine the effect of sclerostin on HDPC differentiation, we compared the odontoblastic differentiation ability between early-passaged SOST and Control groups. As shown in Fig. 3, sclerostin overexpression significantly decreased the odontoblastic differentiation of early-passaged HDPCs as shown by the decreased mineralization nodules formation and a decline in ALP activity along with lower expression levels of odontoblastic differentiation markers such as ALP, OPN, CON and DSPP (Figs. 3A–3C).

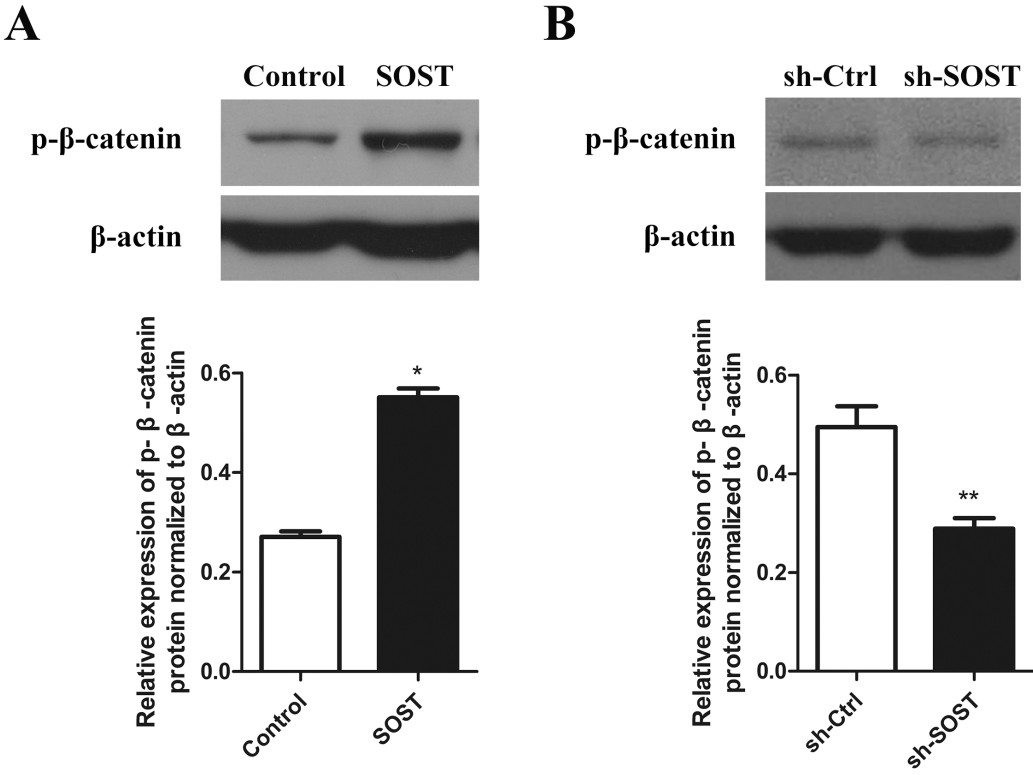

**Figure 4 Effect of sclerostin on Wnt/β-catenin pathway.** (A) Western blot analysis of p-β-catenin expression in early-passaged Control and SOST HDPCs; (B) Western blot analysis of p-β-catenin expression in late-passaged sh-Ctrl and sh-SOST HDPCs. *$P < 0.05$, **$P < 0.01$.

Moreover, knockdown of sclerostin rescued the decreased odontoblastic differentiation of senescent HDPCs (Figs. 3D–3E). These data imply that the increased expression of sclerostin contributed to the impaired odontoblastic differentiation potential of senescent HDPCs.

### The Wnt/β-catenin pathway may be involved in the progress of HDPCs senescence related to higher expression of sclerostin

To clarify the mechanisms underlying sclerostin-related HDPCs senescence, we examined the Wnt/β-catenin pathway activity. Western blot analysis showed that overexpression of sclerostin suppressed the activity of the Wnt/β-catenin pathway by increasing the expression of p-β-catenin and sclerostin knockdown significantly increased the activity of the Wnt/β-catenin pathway (Fig. 4). These results suggest that sclerostin might accelerate senescence of HDPCs in a Wnt signaling pathway-dependent mechanism.

## DISCUSSION

The importance of DPCs in diverse therapeutic application has been increasingly recognized in recent years. However, the impairment of proliferation and differentiation caused by cellular senescence restricts their application in tissue regeneration. So far, little is known about the progress of HDPC senescence. Herein, by sclerostin

overexpression in early-passaged HDPCs and knockdown in late-passaged HDPCs, we showed that sclerostin knockdown is beneficial for the maintenance of the proliferation and odontoblastic differentiation potentials of HDPCs during cellular senescence.

In the present study, we found that there was a significantly higher expression of sclerostin in senescent human dental pulp tissues and senescent HDPCs. These data are consistent with the findings that serum sclerostin levels increase with age, which may contribute to age related bone loss (*Modder et al., 2011*; *Zhang et al., 2016*). While noteworthy, these results do not provide causality. Therefore, SA-β-Gal staining, a biomarker of cellular senescence (*Debacq-Chainiaux et al., 2009*), was performed in early-passaged HDPCs with sclerostin overexpression and in late-passaged HDPCs with sclerostin knockdown to determine whether the higher expression of sclerostin was the cause of HDPC senescence or not. Sclerostin overexpression accelerated HDPC senescence and sclerostin knockdown decreased senescence of HDPCs in vitro. To the best of our knowledge, it was the first study confirming the exact role of sclerostin in cellular senescence. Well-designed studies are now required to determine the role of sclerostin in HDPC senescence in vivo and to identify whether this promoting effect of sclerostin on HDPC senescence is universal in other cells or is cell-type specific.

Senescence of cells is mostly due to activation of G1/S cell cycle arrest proteins (*Itahana, Campisi & Dimri, 2004*). The p53/p21 and p16/retinoblastoma axes are two important pathways in cellular senescence. The p16 protein mediates cell cycle arrest by inhibiting DNA replication via preventing phosphorylation of the retinoblastoma protein (*Rayess, Wang & Srivatsan, 2012*). The p53-mediated response to DNA damage, oxidants and hypoxia induces the expression of p21, leading to cellular senescence by inhibiting the activity of cyclin dependent kinases (*Muthna et al., 2010*; *Tonnessen-Murray, Lozano & Jackson, 2017*). Our study showed that senescent human dental pulp and subculture-induced senescent HDPCs exhibited higher expression levels of p16, p53 and p21. These data are in accordance with previous findings that p16, p53 and p21 are highly expressed in senescent DPCs (*Mas-Bargues et al., 2017*; *Muthna et al., 2010*). Conversely, downregulation of p53–p21 decreased senescence of dental pulp stem cells (*Choi et al., 2012*). Meanwhile, p16 knockdown significantly reduced senescence-related dysfunction of dental pulp mesenchymal stem cells (*Feng et al., 2014*). Inspired by the similar expression profile of p16, p53, p21 and sclerostin during HDPCs senescence, we examined the effect of sclerostin on the expression of p16, p53 and p21 in HDPCs. Sclerostin overexpression significantly increased the expression levels of p16, p53 and p21, whereas, sclerostin knockdown decreased the expression levels of p16, p53 and p21. Although there had previously been no reports about the effect of sclerostin on the expression of p16, p53 and p21, it was believed that senescence acted as a tumor suppressor and sclerostin silence significantly increased the proliferation of osteosarcoma cells by promoting the progress of cell cycle in G1/S phase (*Zou, Zhang & Li, 2017*). Thus, one can conclude that sclerostin may modulate the progression of HDPC senescence via both the p16 and p53 pathways.

Odontoblastic differentiation potential was impaired in senescent HDPCs with decreased expression levels of odontoblastic differentiation markers ALP, OCN, OPN and

DSPP, which play key roles in matrix formation and calcification initialization in bone and teeth (*Bae et al., 2015*; *Chen et al., 2005*; *Kuratate et al., 2008*; *Ma et al., 2009*; *Min et al., 2010*). In this study, we found that sclerostin significantly inhibited odontoblastic differentiation of early-passaged HDPCs with downregulation of odontoblastic markers ALP, OCN, OPN and DSPP. Furthermore, knockdown of sclerostin increased odontoblastic differentiation of subculture-induced senescent HDPCs with higher expression levels of these odontoblastic differentiation markers. These results are in line with the finding that mice DPCs with sclerostin deficiency exhibited enhanced in vitro mineralization (*Mehrazarin et al., 2011*). These results indicate that the increased expression of sclerostin was responsible for impairment of odontoblastic differentiation potential in senescent HDPCs.

It was reported that sclerostin efficiently inhibited Wnt signaling by interrupting the Wnt-Frizzled-LRP5/6 receptor complex via binding with the Wnt co-receptor LRP5/6 (*Semenov, Tamai & Xi, 2005*), leading to upregulated expression of p-β-catenin, thereby preventing the nucleus translocation of stabilized β-catenin and thus downregulating downstream genes expression (*Bae et al., 2015*). Wnt signaling plays an essential role in age-related changes in stem cells. Previous studies found that inhibiting Wnt signaling initiated senescence of glioblastoma cells and human WI38 fibroblasts (*Lambiv et al., 2011*; *Ye et al., 2007*). Furthermore, Wnt1, an agonist of Wnt signaling, rescued the impaired neurogenic differentiation potential of aged dental pulp stem cells (*Feng et al., 2013*). In this study, sclerostin overexpression attenuated the activity of Wnt signaling and sclerostin knockdown activated Wnt signaling. These results indicate that sclerostin might accelerate senescence of HDPCs, in part, by decreasing Wnt signaling. However, the exact mechanism of sclerostin-related HDPC senescence requires further study to elucidate.

## CONCLUSION

Taken together, the higher expression of sclerostin might accelerate HDPC senescence and was responsible for the attenuated proliferative and odontoblastic differentiation potentials in senescent HDPCs via p16 and p53 pathways. Sclerostin may serve as a target to delay the progress of HDPCs senescence.

## ACKNOWLEDGEMENTS

We thank M.D.S. Chufang Liao (School and Hospital of Stomatology, Wuhan University) and M.D.S. Yun Wu (School and Hospital of Stomatology, Wuhan University) for HDPC preparation and for their advice on our work.

### Funding

This work was supported by the National Natural Science Foundation of China (No. 81500888, 81200812, 81300904, 81571011 and 81371170) and Youth Medical Experts' Project of Wuhan. The funders had no role in study design, data collection and analysis, decision to publish, or preparation of the manuscript.

## Grant Disclosures

The following grant information was disclosed by the authors:
National Natural Science Foundation of China: 81500888, 81200812, 81300904, 81571011 and 81371170.
Youth Medical Experts' Project of Wuhan.

## Competing Interests

The authors declare that they have no competing interests.

## Author Contributions

- Yanjing Ou conceived and designed the experiments, performed the experiments, analyzed the data, prepared figures and/or tables, authored or reviewed drafts of the paper, approved the final draft.
- Yi Zhou conceived and designed the experiments, analyzed the data, authored or reviewed drafts of the paper, approved the final draft.
- Shanshan Liang conceived and designed the experiments, contributed reagents/materials/analysis tools, authored or reviewed drafts of the paper, approved the final draft.
- Yining Wang conceived and designed the experiments, authored or reviewed drafts of the paper, approved the final draft.

## Human Ethics

The following information was supplied relating to ethical approvals (i.e., approving body and any reference numbers):

The protocols and procedures were reviewed and approved by the Ethical Committee of the School and Hospital of Stomatology, Wuhan University, China (2015C12).

## Data Availability

The raw data are provided in the Supplemental Files.

## Supplemental Information

Supplemental information for this article can be found online at http://dx.doi.org/10.7717/peerj.5808#supplemental-information.

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
