# Peer review of "Sclerostin promotes human dental pulp cells senescence"

_PeerJ, doi:10.7717/peerj.5808_

## Round 0.1 · original submission · Minor Revisions

Please address the minor comments raised by the reviewers.

·

Basic reporting

no comment

Experimental design

SOST immunochemistry: controls have to be presented at least as supplementary material.
- In Fig 1 the image pixel definition seems a bit low.
The authors should precise with initiale letters the localization of pulp, dentin,
predentin, and odontoblasts on the immuno figures.

- Fig 2: same comment as for fig 1 images, the pixel definition is low.

Validity of the findings

Data is robust, statistically sound, & Controlled
No speculation
Conclusions are well stated, linked to original research question & limited to supporting results.

Additional comments

- The beginning of the introduction, aiming to point out the importance of the dental pulp complexe preservation ,could be ameliorate ( cf pdf notes).
The others aspects ( Sost and its implication in bone formation, aging and dentin regeneration) are very well described.
- Except for the “legal aspect” ( see below “ custom check note as demanded by the editors) , the materiel and method part is particularly tidy and will be really helpful for the readers.
- The results are clearly stated by the authors and answer point by point to their initial questions and objectives
- The discussion provides well-documented informations to the reader and proposes perspectives for further research works.

CUSTOM CHECK:
- Ethical approval and consent form are in Chinese so can’t be verified. An empty form translated in English should be provided for the informed consent as well as for the signed page of the Ethical approval.
- The authors have included a statement in their material and methods part concerning the Declaration of Helsinki and the information consents but no approval reference numbers are mentionned ; neither the name of the granting organization.


-Identifiable info have been removed from all files: OK
- Were the experiments necessary and ethical? YES

·

Basic reporting

Overall, the reporting is good. There are a few minor typos and needs careful copy editing.Eg; Line 67 – Elderly

Experimental design

The manuscript #28635 entitled “Sclerostin promotes human dental pulp cells senescence” by Ou Y et al is appropriate for this journal audience. The authors use primary dental pulp cells and both loss of function (sh RNA) and over-expression to elegantly dissect SOST function.

A few minor issues can be clarified to improve interpretation and clarity of this work.
1. Were the Alizarin studies done at the same time points? This should be explicitly stated.

2. Details of serial sub-cultured are necessary. As per the traditional 3T3 protocols, cell surface area per cell spilt must be carefully documented to account accurately for population doubling.

3. Figure legends and labels are very confusing, they should be redone. Eg; simply label HDPC/PCDH as ‘Control’ and HDPC/SOST as ‘SOST’. This will significantly help the reader follow the data.

Validity of the findings

No comments

Additional comments

The manuscript #28635 entitled “Sclerostin promotes human dental pulp cells senescence” by Ou Y et al is appropriate for this journal audience. The authors use primary dental pulp cells and both loss of function (sh RNA) and over-expression to elegantly dissect SOST function.

A few minor issues can be clarified to improve interpretation and clarity of this work.

1. Were the Alizarin studies done at the same time points? This should be explicitly stated.

2. Details of serial sub-cultured are necessary. As per the traditional 3T3 protocols, cell surface area per cell spilt must be carefully documented to account accurately for population doubling.

3. Figure legends and labels are very confusing, they should be redone. Eg; simply label HDPC/PCDH as ‘Control’ and HDPC/SOST as ‘SOST’. This will significantly help the reader follow the data.

4. A minor typos must be corrected. Eg; Line 67 – Elderly Careful revision of the complete manuscript should be performed.

---

## Round 0.2 · Minor Revisions

Your paper has been re-reviewed by a topic expert who suggests you include the relevant comments about Sclerostin levels in teeth from older patients in your manuscript discussion. Please include brief statements for final acceptance.

Reviewer 3 ·

Basic reporting

The present article passes the requirement for basic reporting topics.

Experimental design

The present article passes the requirement for experimental design topics.

Validity of the findings

The present article passes the requirement for validity topics.

Additional comments

The present study described the role of SOST on human dental pulp cell senescence. The experimental designs are straight forward and several approaches were employed to identify one research question, confirming each other data. Overall manuscript was well-written. However, the discussion on the teeth samples should be added. Authors may obtained teeth from elder group due to the potential extraction from periodontal diseases. It has a possibility that dental pulp of periodontal affected teeth may be compromised. Hence, the observation of SOST expression and p16 p21 p53 expression may also be influence. The discussion on this issue would help to clarify the results.

---

## Round 0.3 · accepted · Accept

The manuscript appears to have been revised suitably - Congratulations!

#